# Identifying reproducible transcription regulator coexpression patterns with single cell transcriptomics

**Alexander Morin**[1,2,3], **Ching Pan Chu**[1,2,3], **Paul Pavlidis**[1,2]*

**1** Michael Smith Laboratories, University of British Columbia, Vancouver, British Columbia, Canada,
**2** Department of Psychiatry, University of British Columbia, Vancouver, British Columbia, Canada,
**3** Graduate Program in Bioinformatics, University of British Columbia, Vancouver, British Columbia, Canada

* paul@msl.ubc.ca

## Abstract

The proliferation of single cell transcriptomics has potentiated our ability to unveil patterns that reflect dynamic cellular processes such as the regulation of gene transcription. In this study, we leverage a broad collection of single cell RNA-seq data to identify the gene partners whose expression is most coordinated with each human and mouse transcription regulator (TR). We assembled 120 human and 103 mouse scRNA-seq datasets from the literature (>28 million cells), constructing a single cell coexpression network for each. We aimed to understand the consistency of TR coexpression profiles across a broad sampling of biological contexts, rather than examine the preservation of context-specific signals. Our workflow therefore explicitly prioritizes the patterns that are most reproducible across cell types. Towards this goal, we characterize the similarity of each TR's coexpression within and across species. We create single cell coexpression rankings for each TR, demonstrating that this aggregated information recovers literature curated targets on par with ChIP-seq data. We then combine the coexpression and ChIP-seq information to identify candidate regulatory interactions supported across methods and species. Finally, we highlight interactions for the important neural TR ASCL1 to demonstrate how our compiled information can be adopted for community use.

## Author Summary

A common way to analyze gene expression (transcriptomics) data is to correlate gene transcript levels across samples for every pair of genes (coexpression). Coordinated expression between genes may imply a shared biological function, though this warrants cautious interpretation given assumptions about cellular processes inferred from RNA abundances alone. Still, coexpression inference is often used to nominate genes whose expression may be controlled by transcription regulators (TRs). The rapid generation of diverse single cell transcriptomics data has unlocked our ability to discover coexpression patterns across individual cells — though these signals are often noisy. Reproducible patterns across studies can help distinguish meaningful biological relationships from

**Data availability statement:** All summarized rankings, scored ChIP-seq experiments, and GO analysis results are made available as R objects in the Borealis data repository (https://doi.org/10.5683/SP3/HJ1B24). The identifiers and associated data links of the analyzed scRNA-seq experiments are found in S1 Table and summaries of the curation benchmark are found in S2–3 Tables. The code to reproduce the analysis is located at https://github.com/PavlidisLab/TR_singlecell.

**Funding:** This work was supported by National Institutes of Health grant MH111099 (https://www.nih.gov/) and Natural Sciences and Engineering Research Council of Canada (NSERC) grant RGPIN-2016-05991 (https://www.nserc-crsng.gc.ca/) and the Canadian Foundation for Innovation Leaders Opportunity Fund held by P.P. A.M. had funding support from the Canadian Institutes of Health Research Canada Graduate Scholarship (CIHR-CGS), Natural Sciences and Engineering Research Council of Canada - Collaborative Research and Training Experience (NSERC-CREATE), and UBC Institute of Mental Health (IMH) Marshall Scholars programs. The funders had no role in study design, data collection and analysis, decision to publish, or preparation of the manuscript.

**Competing interests:** The authors have declared that no competing interests exist.

spurious correlations. We used this study to analyze a broad collection of single cell data spanning numerous tissues in human and mouse to infer global TR coexpression patterns. We aimed to learn which interactions were generally observable, to better potentiate future examinations of reproducible coexpression in specific contexts. We evaluate the predictive performance of these global single cell coexpression rankings using independent gene regulation evidence, and highlight TR-gene pairs that are supported across data modalities as well as species. By disseminating these rankings, we hope that other researchers can extract insight for their own TRs of interest.

## Introduction

The widespread adoption of single cell genomic methodologies, particularly single cell/nucleus RNA sequencing (herein, scRNA-seq), has significantly advanced our ability to characterize dynamic cellular processes. The scale with which scRNA-seq data has been generated has created an unprecedented opportunity to understand the reproducibility of these cellular patterns. This is important because, despite its power, scRNA-seq results in sparse gene transcript counts due to both biological and technical factors [1,2].

Gene regulation is a field that stands to greatly benefit from the single cell era. A primary objective is to map the temporal and context-specific interactions between transcription regulators (TRs) and their target genes. However, understanding the sets of genes regulated by each TR — regardless of context — remains a challenge. Despite the availability of genetic tools, linking TRs to direct gene targets is hindered by multiple factors. These include the cost and difficulty of collecting experimental data implicating direct regulation, such as TR binding information from chromatin immunoprecipitation sequencing (ChIP-seq), and the inherent complexity of the underlying biology [3,4].

Gene coexpression is a traditional and widely adopted approach for predicting TR-target relationships. This analysis is often cast as generating a predicted gene regulatory network, where the strength of covariation between gene transcript levels serves as edge weights [5]. The fundamental assumption is that if a TR protein influences a gene's transcription, the TR gene itself must also be expressed. However, this assumption may be compromised when the dynamic expression of TRs and their targets are uncoupled. Further, this covariation does not implicate a causative directionality (i.e., regulatory influence) between gene pairs. Despite these limitations, coexpression analysis has been extensively applied as a cost-effective and genome-wide strategy to investigate gene regulation and is commonly integrated with other data modalities [6,7].

The emergence of scRNA-seq has made it possible to study coexpression at a finer level of granularity than afforded by bulk tissue, mitigating cell type compositional effects that impact bulk tissue interpretation [8–11]. However, cautious interpretation is still warranted due to the sparsity of scRNA-seq data. Correspondingly, the benefits of a meta-analytic framework [12–14] have been extended to single cell coexpression to tasks such as gene function prediction [1,15] and understanding reproducible patterns in the brain [16–18]. Importantly, these studies typically focused on the preservation of the global coexpression network structure, rather than any specific gene profile.

We drew inspiration from these works and our experience in aggregating ChIP-seq and TR perturbation studies to identify reproducible TR-target interactions [19]. This stemmed from the recognition that the evidence from various lines of gene regulation methods often do not intersect, necessitating comprehensive data compilation [20–24]. In this study, we adopt a "TR-centric" approach towards aggregating single cell coexpression networks, with

the primary goal of learning reproducible TR interactions. Specifically, our focus was to assemble a diverse range of scRNA-seq data to better understand the coexpression range of all measurable TRs in mouse and human. Our key aim was to prioritize the genes that are most frequently coexpressed with each TR, hypothesizing that this prioritization can facilitate the identification of direct TR-target interactions. We further reasoned that this information would help establish expectations for more focused data aggregations.

## Methods

All analyses were performed in the R statistical computing environment (R version 4.2.1 http://www.r-project.org). The associated scripts can be found at https://github.com/PavlidisLab/TR_singlecell.

### Genomic tables

Gene annotations were based on NCBI RefSeq Select (mm10 and hg38) (https://www.ncbi.nlm.nih.gov/refseq/refseq_select/). High-confidence one-to-one orthologous genes were accessed via the DIOPT resource (V9; [25,26]). We kept only genes with a score of at least five that were also reciprocally the best score between mouse and human and excluded genes with more than one match. This resulted in 16,981 orthologous genes. Cytosolic L and S ribosomal genes were obtained from Human Genome Organization (groups 728 and 729; https://www.genenames.org/data/genegroup/#!/group/). This encompassed 89 human genes, which we subset to the 82 genes with a one-to-one mouse ortholog. Transcription regulator identities were acquired from Animal TFDB V4 [27].

### scRNA-seq data acquisition and preprocessing

We focused on datasets with count matrices that had cell identifiers readily matched to author-annotated cell types. This was primarily sourced through two means: 1) From the "Cell x Gene" database (https://cellxgene.cziscience.com/), which has pre-processed and annotated data. When a single submission ("collection") contained multiple downloads (for example, different tissue lineages), we downloaded all and combined them into a single dataset keeping only unique cells. 2) Automated screening followed by human curation of the Gene Expression Omnibus (GEO) database [28]. Here, we preserved the author-annotated cell types, save for when a biologically uninformative delimiter was used (e.g., "Neuron-1" and "Neuron-2"), in which case we collapsed these cell types into one to prevent overly sparse cell-type populations. We further acquired two tissue-panel datasets. The first was downloaded from the Human Protein Atlas [29] (https://www.proteinatlas.org/download/rna_single_cell_read_count.zip, June 2023), covering 31 tissue-specific datasets which we collapsed into a single dataset and thus treated as a single network. Similarly, we downloaded each of 20 tissue datasets from the Tabula Muris Consortium [30] (https://figshare.com/articles/dataset/Robject_files_for_tissues_processed_by_Seurat/5821263; July 2023), which were also combined as one dataset.

Following the advice of the Harvard Chan Bioinformatics Core (https://hbctraining.github.io/scRNA-seq_online/lessons/04_SC_quality_control.html), we uniformly applied relatively lenient filtering rules for all datasets. We required a minimum cell count of 500 UMI (or equivalent) and 250 expressed genes, and a ratio of the $\log_{10}$ count of genes over $\log_{10}$ UMI counts greater than 0.8 for all experiments, save for SMART-seq assays, where the cutoff was relaxed to 0.6 as this technology can result in greater read depth for select genes [31]. We applied standard CPM library normalization on the raw counts of all datasets (Seurat V4.1.1 NormalizeData "RC"), having observed that the log transformation in other normalization schemes resulted in elevated correlation reproducibility in our null comparisons.

## scRNA-seq network construction

Aggregate single cell coexpression networks were constructed as described by Crow et al. [1]. Every dataset consists of a gene by cell normalized counts matrix, where each cell is associated to an annotated cell type. We fix genes to the RefSeq Select protein coding genes, setting unreported genes to counts of 0. This was done so that every resulting network had equal dimensionality.

For a given dataset, we performed the following steps for each cell type:

1. Subset the count matrix to only cells of the current cell type.

2. Set genes with non-zero counts in fewer than 20 cells to NA.

3. Calculate the gene-gene Pearson's correlation matrix.

4. Set NA correlations resulting from NA counts to 0.

5. Make the correlation matrix triangular to prevent double ranking symmetric elements.

6. Rank the entire correlation matrix jointly, using the minimum ties method.

The resulting rank matrices across cell types were then summed into one matrix that was re-ranked and standardized into the range [0, 1] by dividing each element by the maximum rank. Higher values correspond to consistently positive coexpressed gene pairs, and values closer to 0 represent more consistently negative pairs. Step 2 is applied to ensure noisy coexpression values are not calculated from overly sparse populations, as recommended by Ballouz et al. [14]. The zero imputation in Step 4 is to ensure the ranking procedure includes non-measured genes, placing them in between positive and negative correlations. Thus, each dataset is represented as a single gene by gene matrix of coexpression scores aggregated across all labeled cell types. A gene profile refers to a single gene vector (such as a TR gene) from a single matrix; a set of profiles is the collection of profiles extracted from the experiments that measured the given gene.

## Gene profile similarity

Coexpression profiles from any one dataset may not have a full complement of measured genes and thus contain tied ranks corresponding to missing values in our framework. Consequently, metrics of similarity that compare all of two lists, such as Spearman's correlation, are inappropriate and so we focused on the agreement of the Top and Bottom K genes between profiles. We calculated various set overlap metrics between lists and, finding our conclusions to be consistent, opted for the interpretability of reporting the size of the $Top_K$ and $Bottom_K$ overlaps. We restrict our reporting to TRs that were measured in at least five datasets.

For each TR, we calculated the pairwise similarities among its set of profiles. Averaging these similarity metrics was used to infer the consistency of a TR's coexpression profile across datasets. This process was also applied to each of the 82 ribosomal genes to provide a comparison with a set of genes known to be coexpressed. To generate a null comparison, a random TR was selected from each network to create a set of shuffled profiles, and pairwise similarities were calculated and averaged as above. This process was repeated 1000 times, generating a null distribution of average pairwise similarities. A TR with an average similarity greater than any of the 1000 nulls has an empirical *p-value* < 0.001.

## Aggregating TR profiles and the effect of gene measurement sparsity

To prioritize the gene partners most commonly coexpressed with each gene, we averaged the set of rank-standardized profiles for the given gene into one aggregate profile. As each dataset-level profile had variable gene measurement, there was variable delineation between

the positive coexpression values, the non-measured gene pair ties, and negative coexpression values. Therefore, for a given gene's set of profiles, we imputed all tied values to the median value before averaging, standardizing the values of non-measured gene pairs. A schematic is shown in S1C Fig.

## Gene set enrichment

For each aggregate profile, we performed GO enrichment analysis of "biological process" terms with the "precRecall" R implementation of ermineJ (https://github.com/PavlidisLab/ermineR) [32], using the aggregate values as scores. This approach considers the full scored list to find enriched terms but places greater emphasis on the top of the gene list. We analyzed 3,284 terms that had 20-200 genes and set the false discovery rate at 0.05 for considering terms significant. For the orthologous coexpression rankings we used human genes to map GO annotations.

## ChIP-seq data acquisition and summarization

All ChIP-seq data was downloaded from the Unibind database [33] (https://unibind.uio.no/downloads/; September 2022). For every TR experiment, we scored gene binding intensity using the same approach as in Morin et al. [19], using a continuous scoring function [34] (detailed in S1 Text). To generate an aggregate binding profile, we averaged the gene binding vectors specific to each TR. A "consensus" list of ASCL1 bound regions consisted of the union of all its peaks across ASCL1 Chip-Seq datasets (detailed in S1 Text).

## Literature curation evaluation

TR-target interactions supported by low-throughput experimental evidence were collected from our prior study [35], which compiled information from other resources (see S1 Text for details) and then significantly expanded upon neurologically-relevant TRs. Since Chu et al. [35] was published, we have further expanded this collection, to a total of 27,629 experiments encompassing 772 TRs and 5,899 gene targets. We then used each TR's aggregate profile's ranking as a score and its curated targets as labels, calculating AUC metrics (AUPRC: area under the precision recall curve and AUROC: area under the receiver operator curve) using the ROCR package (V1.0-11) [36]. To generate a null comparison for each TR, we randomly sampled from the entire literature curation corpus a number of targets equal to the count of curated targets for the given TR, and calculated AUCs using the TR's aggregate profile as a score and the shuffled targets as labels. This process was repeated 1000 times to generate a null distribution of AUC values. The observed AUCs (using the TR's true curated targets) were then compared to this distribution of null AUCs. A quantile of 1 means that the observed AUC was better than every single null AUC (empirical *p-value* < 0.001). We restrict our reporting to TRs that had at least five curated targets.

## Cross-species coexpression profile comparison

There were 1,246 TRs with a one-to-one orthologous match between mouse and human that were also measured in at least 5 datasets in each species. For each of these TRs, we subset their aggregate profiles to the 16,981 orthologous genes. Each orthologous TR thus has a mouse and human aggregate profile, generated separately across the respective species' datasets. To generate a consensus orthologous profile for each TR, we took the rank product between its human and mouse aggregate profiles. To compare ortholog aggregate profiles, we calculated Spearman's correlation and TopK and BottomK overlaps. Null comparisons were generated in a manner consistent with the individual profile comparison: similarities were calculated between randomly shuffled aggregate profiles between species over 1000 iterations.

To learn the specificity of a TR's aggregate coexpression profile with its matched ortholog in the reciprocal species, we combined the framework applied in this study with prior studies examining the conservation of coexpression [17,37]. For each TR in a species, we selected the given TR's top 200 coexpressed partners ($Top_{200}$). We next calculated the overlap of this gene set with the $Top_{200}$ gene sets of each of the 1,246 TRs in the other species. We then treated the mismatched (non-orthologous) overlaps as a distribution and represented the matched (ortholog) TR's $Top_{200}$ as a quantile with respect to this distribution. We refer to this quantile as the *Ortholog retrieval score*. A score of 1 means that the given TR's ortholog shared more top coexpressed partners than any other TR in the other species. This procedure was then repeated for the reciprocal species. The result is a pair of *Ortholog retrieval scores* for each TR: how well a human TR's aggregate profile recovered its mouse ortholog relative to all other mouse TRs (human in mouse), and the recovery of the mouse ranking across human TRs (mouse in human).

## Integrating coexpression and ChIP-seq profiles

For TRs with ChIP-seq data, we took the rank product of the TR's aggregate coexpression profile and its aggregate binding profile, re-ranking the result [19,38,39]. This orders genes by placing equal weight on their (positive) coexpression evidence and their binding evidence. We further report a second prioritization scheme for each TR, categorizing genes based on a cut-off of the rankings for both data types and species:

1. Stringent: Required a gene's presence in the Top 500 of both coexpression and binding in both species (orthologous genes only).

2. Elevated: Genes needed to make the Top 500 cut-off for both data types in one species and in one data type for the other species (orthologous genes only).

3. Species-specific: Top 500 cut-off for both data types in one species only. Notably, this may include genes absent from the one-to-one orthologous set, or TRs that had ChIP-seq data in one species only. Consequently, this tier had greater coverage than the others.

4. Mixed-species: Allowed genes ranked in the Top 500 in both data types, but each in only one species (orthologous genes only).

## Results

### Assembling a broad corpus of single cell RNA-seq data

To establish a diverse range of biological contexts for constructing single cell coexpression networks, we acquired scRNA-seq data from public resources (Methods). Our focus was strictly on datasets that included author-annotated cell type labels in the metadata, and all identified datasets underwent consistent preprocessing. In total, we analyzed 120 human datasets and 103 mouse datasets (Fig 1A; Metadata in S1 Table). This corpus spans a wide range of biological contexts, scRNA-seq technologies, and counts of assayed cells. After filtering, the median human dataset measured 15,341 protein coding genes across 74,148 cells and 14 cell types; in mouse 13,996 genes across 36,755 cells and 12 cell types (Figs 1B, S1A and S1B). There was appreciable spread in these counts, with tissue atlas studies typically exhibiting the broadest coverage. The complete dataset is over 2.8 x$10^7$ cells.

### Constructing single cell coexpression networks

We constructed aggregated single cell coexpression networks for each dataset using the approach outlined by Crow et al. [1] (Methods). In brief, this entails generating a

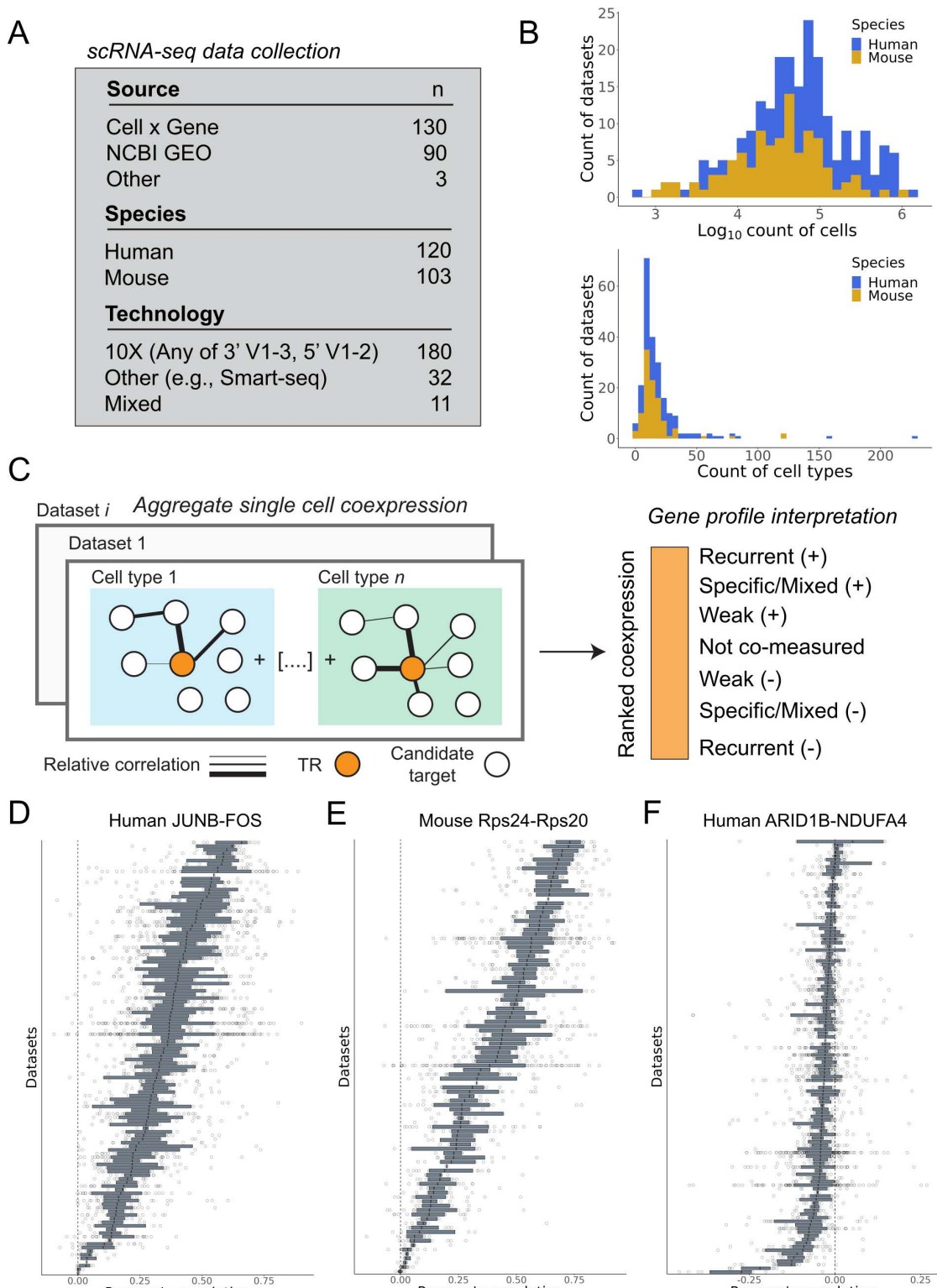

**Fig 1. Overview of study design.** (A) Counts of datasets by source, technology, and species. (B) Top panel: Counts of cells across the dataset corpus. Bottom panel: Counts of cell types. (C) Schematic of the single cell coexpression aggregation framework and the interpretation of an individual gene coexpression profile. (D, E) Examples of the most reproducible positively coexpressed gene pairs. Each bar represents a dataset/network, and each point represents the gene pair's correlation in a cell type within the dataset. (F) Example of one of the most reproducibly negative coexpression gene pairs.

gene-by-gene correlation matrix for each cell type within a dataset, ranking each cell type correlation matrix, and consolidating them into a single network per dataset (Fig 1C). Notably, unlike in Harris et al. [16], where information was consolidated across datasets for a single cell type, we first aggregate across cell types within a dataset before aggregating across datasets. In doing so, we explicitly prioritize signals shared across cell types. This strategy also minimizes effects due to expression differences between cell types, which we consider a separate question from "within cell" regulatory interactions [10].

This procedure aims to rank coexpression partners, as illustrated in Fig 1C, by ordering from "top" to "bottom": consistently high positive interactions across cell types; mixed/specific positive interactions; weak-to-no coexpression; non-measured gene pairs; and then the increasingly most reproducibly negative coexpressed pairs. From this network, it is possible to extract a single gene column (herein, gene profile), such as for a TR, with the relative ordering reflecting the strength of its aggregate transcript covariation with all other genes.

While the focus of this study is on TRs, we first examined the globally most consistent coexpressed gene pairs (Fig 1D–F). Top examples include TRs that dimerize to form the pleiotropic AP-1 complex, such as JUNB and FOS, as well as members of the ribosomal complex. Given the known biological coexpression of ribosomal genes [40], we use a set of 82 large (L) and small (S) ribosomal genes that are highly conserved between mouse and human as a positive control when examining TR-gene coexpression in the following analyses (Methods). We also show one of the most consistently negative coexpressed TR-gene pairs in human. Aligning with our prior observations [12], the magnitudes of these values are smaller and less consistent than the positive coexpression profiles, contributing to the complexity in identifying repressive interactions (discussed in S1 Text).

## Similarity of TR-target profiles

Before prioritizing reproducible TR-gene interactions, we examined the concordance of the TR coexpression profiles between datasets. We expected that distinct profiles generated for the same TR and similar contexts would have elevated similarity relative to mismatched contexts or gene profiles. At the same time, the underlying data we used was from differing cell types, as datasets could be from different tissues. While we expected this would affect the degree of similarity, a total absence of overlap between profiles would raise questions about the efficacy of our framework in finding reproducible interactions.

We report here on the size of the overlap ($K$) of the top positively coexpressed ($\text{Top}_K$) genes between each pair of gene profiles (negative coexpression is discussed in S1 Text). We examined a range of $K$, from 200 — approximately the top 1% of protein coding genes — to 1000, finding that our main conclusions were robust to this cut-off. To contextualize the similarity between TR profiles, we generated null similarities, iteratively sampling TRs across datasets and calculating the overlap of the shuffled TR profiles. We also report the similarity of the set of 82 L/S ribosomal genes.

First, for each TR we pairwise compared its profiles across studies. As expected, the most similar pairs were supported by datasets investigating similar biological contexts. For example, the best pairing in human ($\text{Top}_{200} = 163/200$) was between *FOXM1* profiles from two studies that both assayed the developing human intestine [41,42]. The highest Mouse $\text{Top}_{200}$ (150/200) was associated with E2f8, derived from two studies of the blood-brain barrier [43,44]. The magnitude of the best ribosomal gene pairs was comparable: the best global human ribosomal pairing ($\text{Top}_{200} = 161/200$) belonged to *RPS13*, originating from two immune cell studies [45,46].

While these observations support the ability to find consistent coexpression patterns within pairs of similar contexts, our ultimate aim was to combine information across contexts. Seeking a more global summary of TR profile overlap, we calculated the mean $Top_{200}$ overlap for each TR profile across all unique pairs of networks measuring the TR. We again use the similarities from the pairs of randomly sampled TRs and the 82 ribosomal genes as reference.

In Fig 2A and 2B, we show the average $Top_{200}$ of shuffled TR pairs across 1000 iterations. The typical null sample had an average $Top_{200}$ value of 2.7/200 in human and 2.6/200 in mouse. The ribosomal genes, approximating an empirical "upper bound," averaged 61/200 in human and 44/200 in mouse. The distribution of average $Top_{200}$ values was highly skewed for TRs, with 67% of human TRs and 68% of mouse TRs having an average $Top_{200}$ value greater than the maximum value achieved across all of the null samples (empirical *p-value* < 0.001; represented as red lines in Fig 2A and 2B). And while the best individual ribosomal data pairs were equivalent in overlap size compared to the best individual TR pairs, ribosomal genes typically had a much greater average $Top_{200}$ than even the best TR. This underscores the unusual uniformity of ribosomal protein gene coexpression across distinct cellular contexts — it is an outlier. A similar comparison for the $Bottom_{200}$ is provided in S2A–D Fig.

TRs with the highest mean $Top_{200}$ values, indicative of the most consistent positive coexpression profiles across studies, were often associated with fundamental cellular housekeeping processes. For example, *E2F8* led in human (mean $Top_{200}$ 40.4/200), with mouse *E2f8* similarly having one of the most consistent profiles (Fig 2A and 2B). The E2F family are well characterized regulators of the cell cycle [47], and other E2F members also ranked high in both species, as did regulators involved in early transcriptional response to environmental signals, such as AP-1 complex members *FOS* and *JUN*. In mouse, the highest mean $Top_{200}$ belonged to *Mxd3*, a MYC-antagonist whose human ortholog also had elevated similarity. More broadly, there was appreciable correlation between human and mouse orthologous TRs (Methods) in the consistency of their positive and negative coexpression profiles (S3A and S3B Fig).

TRs with context-restricted activity might be expected to exhibit relatively low cross-dataset similarity in our broad corpus. However, this is not necessarily the case. For example, the neural regulator NEUROD6 [48] had one of the most consistent TR profiles in human (mean $Top_{200}$ rank 44th out of 1,605 TRs), despite being only measured in 22 of 120 datasets. This shows that restricted expression does not preclude the identification of reproducible patterns. In contrast, human PAX6 — necessary for the development and function of several nervous and pancreatic tissues [49,50] — had a mean $Top_{200}$ value marginally above the null, improving slightly at K=1000 (S2E and S2F Fig). Although PAX6 can also be described as a context-restricted, it was detected in 85 of 120 datasets, suggesting greater heterogeneity in its coexpression profiles compared to NEUROD6.

## Ranking aggregated coexpression to prioritize TR-target candidates

The preceding section demonstrated that similar TR profiles could be identified across this biologically heterogeneous corpus, supporting the potential to find reproducibly coexpressed gene pairs. We thus turned to our primary aim of prioritizing these consistent interactions, generating a unified gene ranking for each TR using all compiled data. This process involves aggregating information at two levels: first, across cell types *within* a dataset (as in the previous section; Fig 1C), and then, for each TR, aggregating their profiles *across* datasets (Methods and S1C Fig). This approach aims to maintain the interpretability of an aggregate profile

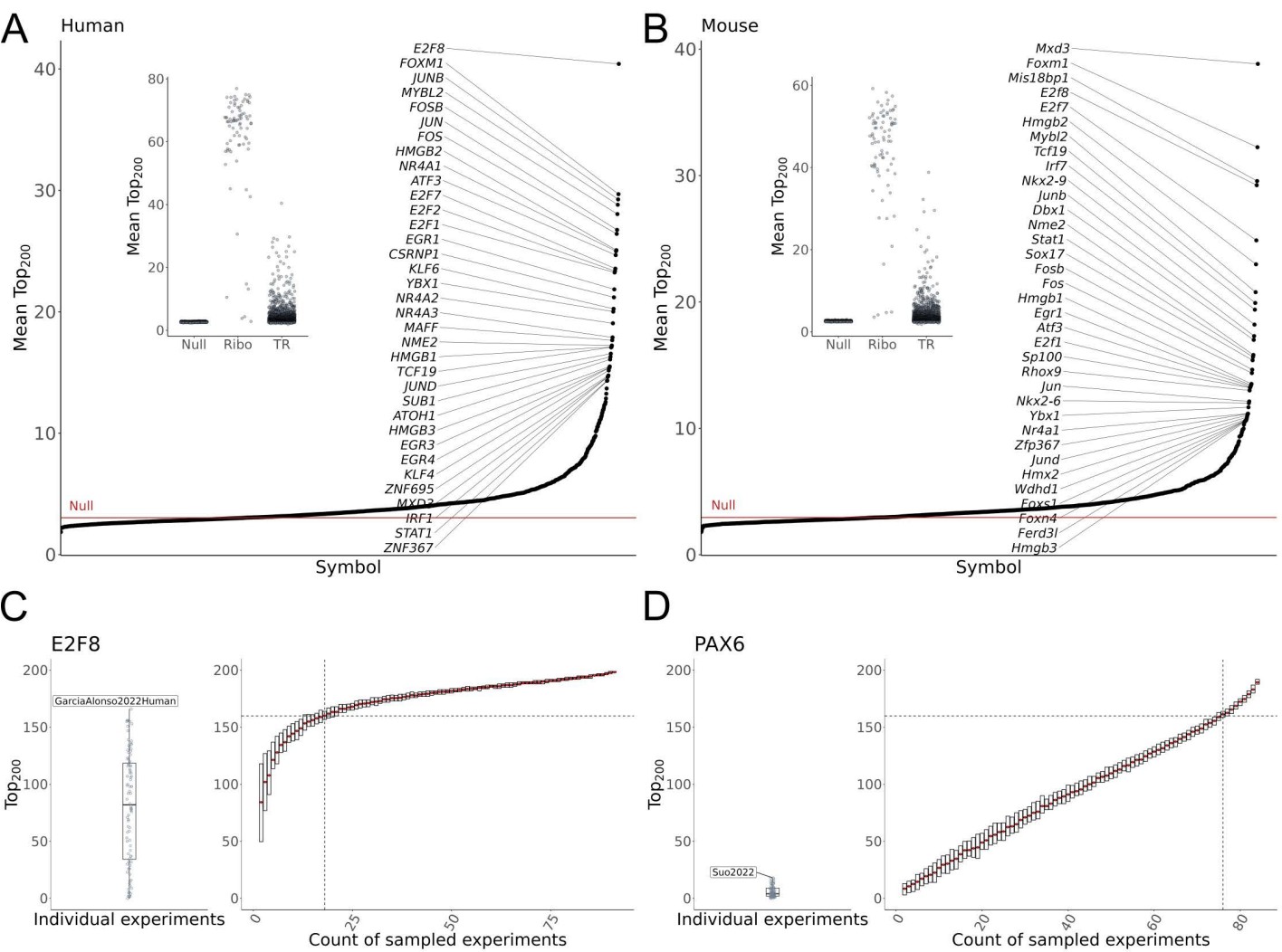

**Fig 2. Similarity of TR profiles.** (A) Inset: distribution of the mean Top$_{200}$ overlaps for the null background, 82 ribosomal genes, and 1,605 human TRs. The null was generated through 1000 iterations of sampling one TR profile from each of 120 human datasets and calculating the average size of the Top$_{200}$ overlap between every pair of sampled profiles. The ribosomal genes represent a "base case" scenario. Main: The average Top$_{200}$ overlap of all human TRs, with the red line indicating the best null overlap. (B) Same as in A, save for 103 mouse experiments and 1,484 TRs. (C,D) Saturation analysis of global TR profiles for human (C) E2F8 and (D) PAX6. Left panels show the spread of Top$_{200}$ overlaps between individual dataset profiles and the global E2F8 and PAX6 profiles. Right panels show the spread of overlaps when iteratively subsampling and aggregating datasets at increasing steps. Dotted lines indicate the average number of sampled datasets required to reach 80% of the global profile. E2F8 recovers its global profile with relatively fewer datasets than does PAX6.

relative to a profile from an individual network (Fig 1C): the extremes represent the most consistent positive and negative correlations, while the middle of the list encompasses weak and non-measured coexpression gene pairs.

As before, we used the set of ribosomal genes to validate that our aggregation workflow prioritized known biological coexpression (S1 Text; S4A and S4B Fig). We next performed GO biological process enrichment on all aggregate profiles (S4C Fig), finding that most TRs (91% human, 86% mouse) were associated with at least one term (FDR 0.05). E2F8 coexpression partners were enriched for multiple terms relating to cytokinesis and chromosomal organization, as expected for its known role in these processes [47]. We also frequently observed that terms affiliated with tissue-specific processes were enriched for TRs implicated in those tissues. Examples include glial development and myelination terms for the oligodendrocyte TRs

OLIG1/2 [51], neuronal synaptic functionality for the aforementioned NEUROD6 [48], leuko-cyte and cytokine processes for IRF8 [52], and hematopoietic terms for the erythroid GATA1 [53]. Some tissue-selective TRs were enriched for more general regulator terms (e.g., "cell fate commitment" for mouse Pax6) or had disparate tissue-specific terms (e.g., "regulation of osteoblast differentiation" and "regulation of neuron differentiation" for SOX4), potentially reflecting data heterogeneity. While GO is an imperfect resource, these results agree with our other observations that our analysis yields biologically-relevant signals.

We examined the relationship between the aggregated global TR profiles and the constitu-ent datasets through two analyses. First, we assessed how well individual experiments aligned with the global profiles to identify potential biases (S1 Text). As shown in S2H–L Fig, datasets with the highest agreement were large studies of broad tissues using the 10X Chromium plat-form, though consistencies between platforms were still observed (S2G Fig).

Second, we performed a saturation analysis to determine how many datasets are needed, on average, to recover each TR's global profile (≥80% overlap in Top$_{200}$ genes). By iteratively subsampling and aggregating datasets, we evaluated the convergence of sampled TR pro-files to the global set. For example, E2F8 (Fig 2C) required an average of 18 of 92 datasets to reach saturation, while PAX6 (Fig 2D) showed a linear trend, indicating saturation has not yet been achieved. These results suggest future work is needed to explore not only replicable context-specific patterns for TRs such as PAX6, but also the extent to which globally consis-tent partners can be found when using more data.

## Recovery of literature-curated TR-target interactions

Equipped with a unified single cell coexpression profile for each human and mouse TR, we aimed to assess their concordance with an orthogonal line of regulation evidence. While coex-pression is expected to prioritize both direct and indirect regulatory interactions (the latter we would consider false positives), the rankings should still demonstrate a greater ability to recover true direct interactions relative to a null expectation.

In a previous study [19], we evaluated the utility of aggregating TR perturbation and ChIP-seq experiments, using literature-curated low-throughput interactions as positive labels and calculating area under the curve (AUC) metrics [23,54]. We applied the same framework here, using curated TR-target interactions we have collected (Chu et al. [35], since expanded) and assembled from other resources (see S1 Text for further discussion). We considered TRs that had a minimum of five curated targets, resulting in 451 TRs analyzed in human (median count of curated targets = 18) and 434 in mouse (median count = 17).

We first examined the effectiveness of the aggregate profiles in recovering curated targets relative to the individual TR profiles that compose each aggregate. On average, the aggregate profiles outperformed (better prioritized curated targets) the expected AUC value from an individual profile (S5A Fig). Therefore, aggregating the coexpression networks typically main-tains or improves performance on this benchmark.

Next, we evaluated the efficacy of the coexpression rankings in recovering curated targets relative to a null distribution of AUCs (*Quant_coexpression)*. While the raw AUC values were typically better than random (Figs 3B and S5B), we report the quantile of the observed value relative to a null to standardize the comparison across TRs (discussed in S1 Text). This null was created by size-matching and randomly sampling from the pool of curated targets from the entire literature-curation corpus. The latter helps account for biases in the coverage of targets in the low-throughput literature. A *Quant_coexpression* value of 1 indicates that an aggregate profile outperformed every null sample.

ASCL1 is provided as an example of this procedure for one TR in Fig 3A. As illustrated in Fig 3C, the coexpression aggregates consistently exceeded the null AUCs, reflected by a

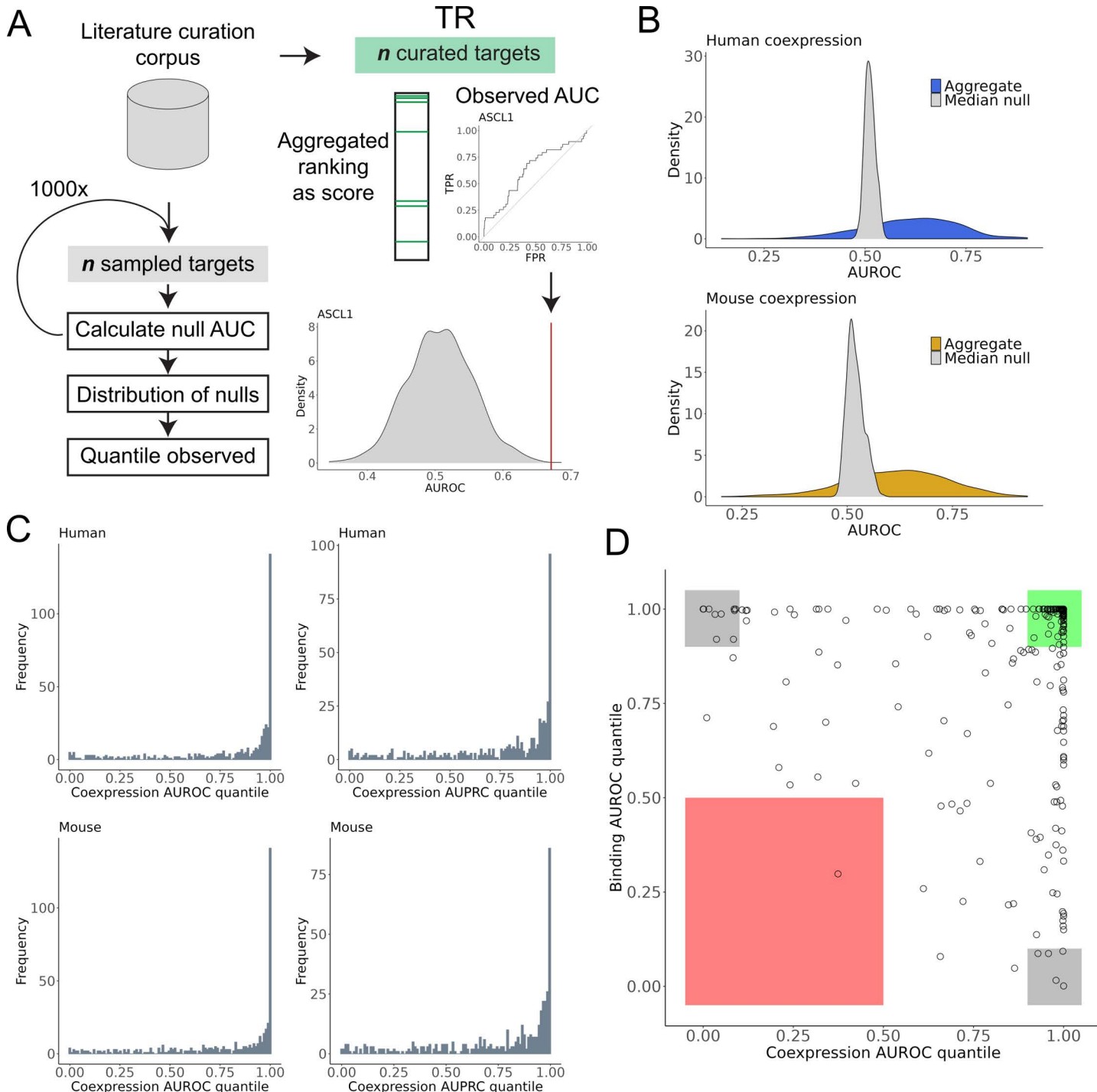

**Fig 3. Recovery of literature curated targets by aggregate rankings.** (A) Schematic of literature curation evaluation. (B) Distributions of the observed AUROCs for 451 human and 434 mouse aggregate TR coexpression profiles, along with the distribution of the median null AUROCs generated for each profile. (C) Histograms of the AUROC and AUPRC coexpression quantiles for human and mouse. (D) Scatter plot of the AUROC quantiles for the coexpression and binding profiles of 253 human TRs that had binding data and at least five curated targets. Green box indicates TRs for which both genomic methods were effective in the benchmark, grey box for only one method, and red box for neither method being effective.

median AUROC *Quant_coexpression* of 0.95 in human and 0.93 in mouse. The pile-up of quantiles near or equal to 1 indicates that, while not universal, a majority of TR single cell coexpression rankings excelled in prioritizing matched curated targets over randomly sampled targets. These observations strongly suggest that these aggregate rankings are capable of prioritizing regulatory interactions that were identified through targeted biochemical assays.

To further contextualize these performances, we conducted a similar null AUC analysis, this time using aggregate ChIP-seq signals. In brief, we applied the same approach as in Morin et al. [19], scoring gene-level binding intensity for each ChIP-seq experiment, then averaging these signals within each TR's set of experiments to create a single unified ranking of gene binding for each TR. In total, we considered 4,115 human experiments for 253 TRs and 3,564 mouse experiments for 241 TRs from the Unibind database [33] (Methods) that had at least five curated targets. As with the aggregate coexpression signal, we compared the unified binding ranking's ability to recover TR-specific curated targets relative to a null of sampled targets (*Quant_binding*).

We anticipated that TR ChIP-seq, as a more direct form of regulatory inference, might outperform coexpression [23]. However, in our hands the aggregate binding evidence was on par with single cell coexpression in its ability to predict known targets (S5C Fig), further motivating integration of both data types. Supporting this, integrating the coexpression and binding rankings for available TRs typically led to elevated performance in the benchmark (S5D Fig).

Among TRs with both binding and coexpression data, many performed well in the benchmark for both data types separately, as demonstrated for human TRs in Fig 3D. In human, 134 of 253 (53%) TRs had AUCs (AUPRC or AUROC) *Quant_binding* > 0.9 and *Quant_coexpression* > 0.9; in mouse 126 of 241 (52%). This signifies that, for these specific regulators, aggregated coexpression and binding profiles both effectively prioritize curated TR targets relative to sampled targets. This alignment highlights TRs whose activity may be more readily identified through distinct data modalities. Further, of the TRs performant in both lines of evidence, more than half did so in both species (human 83 of 134, mouse 83 of 126), suggesting convergence of evidence across not only experiments, but also species.

This agreement of evidence encompassed broadly active TRs, such as those involved in the AP-1 complex. However, it also included more specialized factors, such as the neuronal-specifying ASCL1, and the aforementioned PAX6. This suggests that, even though the average overlap of PAX6 profiles was weak (Figs 2D, S2E and S2F), there was still a consensus of recurrent curated PAX6 targets within these smaller intersects. We also find cases where only one data type was performant. LEF1, for example, had an AUROC *Quant_coexpression* value of 1 in both species but a *Quant_binding* value of 0 and 0.22 in human and mouse, respectively.

Finally, because negative expression correlations might be of interest for identifying repressive interactions, we conducted an analysis of the reproducibility and performance of the relations predicted from the bottom of the rankings. We found that for some TRs, negative correlations performed better than positive correlations in the benchmark, though this was the exception (S5B Fig). This suggests that for some TRs, repressive activity might be inferable from coexpression (see S1 Text for discussion; S2 and S3 Tables for performance values for each TR).

## Identification of conserved interactions

It has been observed that, despite the high evolutionary turnover of regulatory DNA sequences, TR-target relations exhibit relatively high conservation [55], with coexpression providing an attractive means to nominate common and divergent interactions [17,56,57].

Here, our aim was to identify the extent to which individual TR aggregate coexpression profiles were preserved between mouse and human, focusing on orthologous genes (Methods). A meta-analytic comparison of TR single cell coexpression profiles between these two species is lacking, and we reasoned that evidence of conservation using this global data corpus would provide future support for studies that focus on specific TR patterns in a more focused context.

Fig 4A demonstrates the similarity distributions between ortholog aggregate coexpression profiles, overlaid with the median observed and shuffled null values. Although there was appreciable spread in these similarity metrics, most orthologs shared more similarity in their profiles than would be expected from shuffled TRs, suggestive of conserved TR coexpression. While there are TRs that agree less well between species, we are cautious in interpreting this as species-specific regulatory rewiring, given the relatively modest effect size and the absence of an exact match in cellular contexts covered across both species.

Given our emphasis on reproducible interactions, we focused on the overlap at the extremes of these species rankings (Figs 4B, 4C and S6C). To quantify the specificity of this overlap, we applied a slightly modified framework of the $Top_K$ overlap used in this study, consistent with prior studies [17,37] (Methods) and illustrated in Fig 4B. The result is a pair of ortholog retrieval scores for each TR: how well a human TR's ranking recovered its mouse ortholog relative to all other mouse TRs (human in mouse), and the recovery of the mouse ranking across human TRs (mouse in human), with a value of 1 indicating perfect retrieval.

As demonstrated in Fig 4C, there was considerable preservation of single cell aggregate TR coexpression profiles between mouse and human. The median ortholog retrieval score for human was 0.969, with 175/1,246 (14%) TRs having a perfect value of 1; in mouse these values were 0.973 and 172/1,246 (14%), respectively. These relative values correspond to a median $Top_{200}$ overlap of 14 genes, with FOXM1 and HMGB2 each having a maximal $Top_{200}$ of 149 genes (Fig 4A). While the most conserved TRs (by $Top_{200}$ overlap) were led by regulators of housekeeping processes such as cell division, we also observed this preservation among more specific TRs, such as the aforementioned NEUROD6 (human in mouse and mouse in human = 1, $Top_{200}$ = 50). Logically, many of these highly preserved TRs also had similar profiles across datasets within species (as shown in Fig 2A and 2B), and those that were weakly preserved generally lacked consistency within species (S6A and S6B Fig). These findings collectively contribute to characterizing the extent to which each TR can be defined by a set of coexpressed gene partners, facilitating inferences into their biological roles.

In Fig 4C we illustrate this overlap procedure for ASCL1, an essential pioneer nervous system regulator that is also relevant to cancer [58]. Of the 200 genes that were most consistently coexpressed with human ASCL1, 32 of their mouse orthologs were also in the mouse Ascl1 $Top_{200}$ set. This marked the largest overlap human ASCL1 had with any mouse TR (human in mouse = 1). In the reciprocal comparison, where mouse Ascl1 was queried against all human TRs, human ASCL1 ranked 30th (mouse in human = 0.98). The 29 human TRs with a greater overlap with mouse Ascl1 did not have a sizable overlap in the reciprocal comparison, save for HMGB3. Conversely, TRs other than ASCL1 with elevated overlap across species included the ASCL1 curated targets INSM1, HES6, and DLX5 [59–61]. Other TRs are well-characterized for operating in a regulatory network with ASCL1 — though not necessarily as direct downstream targets — such as DLX1/2/6, GSX1/2, SP8, and OLIG2 [62–66]. GO enrichment of the top ASCL1 coexpressed gene partners using information from both species returned numerous terms that are consistent with ASCL1's role in brain development (Fig 4D).

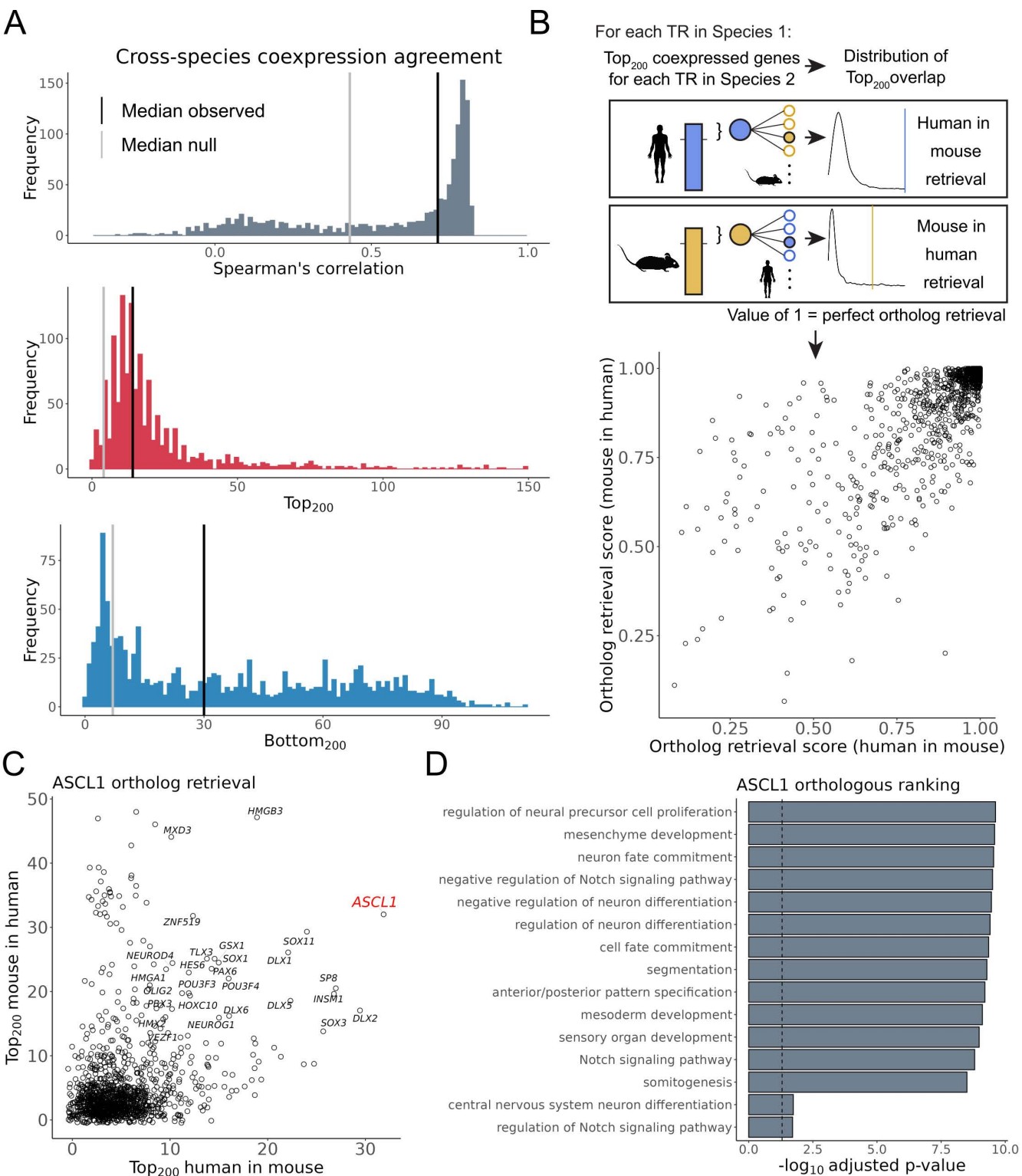

**Fig 4. Preservation of mouse and human single cell coexpression profiles.** (A) Distribution of coexpression agreement between the aggregate single cell coexpression profiles of 1,246 orthologous TRs. Black lines indicate the median value for the TRs, grey lines indicate the median of null values generated by shuffling pairs of orthologous TRs. (B) Top: Schematic of the ortholog retrieval workflow, adapted from Suresh et al., 2023 [17]. Bottom: Scatterplot of the resulting ortholog retrieval scores (C) Scatter plot of the ASCL1 $Top_{200}$ overlaps. (D) The top 15 GO terms when combining the human and mouse top ASCL1 coexpressed gene partners.

## Combining single cell coexpression and aggregated binding reveals numerous reproducible interactions

Up to this point, we have presented evidence supporting the existence of recurrent single cell TR-gene coexpression patterns within (Fig 2) and across species (Fig 4), demonstrating that this information can prioritize curated experimental interactions (Fig 3). One of our primary motivations is to prioritize the direct gene targets of TRs [19]. However, the correlation of TR-gene transcripts serves as an indirect form of gene regulation evidence — it does not confer information about the causative directionality of this covariation. We thus now turn to identifying interactions corroborated by TR binding evidence, using the same aggregated Unibind ChIP-seq data examined in the literature curation evaluation. We reasoned that, as in our earlier work, knowledge of binding can help focus attention on expression patterns more likely to reflect direct regulatory relations.

We present two straightforward strategies for prioritizing reproducible interactions, acknowledging the use of relatively arbitrary cut-offs for the sake of reporting. All summarized rankings are made available for researchers interested in conducting their own exploration. We first combined the single cell coexpression and binding profiles into a final ordered ranking for TRs with ChIP-seq data, using the common rank product summary [19,38,39]. This was done separately for each species (317 TRs in human, 305 in mouse), as well as across species for orthologous TRs with available data (216 TRs). This establishes convenient lists that order the protein coding genes most associated with each TR based on their aggregated single cell coexpression and binding profiles.

Recognizing that a gene may be prioritized (have a better rank product) if ranked exceptionally well in one data type or species only, we introduce a second scheme for more balanced consideration across lines of evidence. For each TR, genes are categorized into tiers by their status across the rankings, as illustrated in the inset of Fig 5A. This collection provides examples of regulatory interactions supported by both binding and single cell coexpression evidence.

Fig 5A shows the counts of unique orthologous interactions gained in each tier of evidence for the available TRs. The Stringent level, representing the most reproducible interactions across both species and genomic methods, contains 545 TR-gene pairs corresponding to 101 TRs and 357 unique genes. The TRs with the largest Stringent collection featured multiple AP-1 members, led by FOSL1 with 29 genes, along with immunity TRs such as STAT1, STAT2, and IRF1. More specialized TRs also had among the largest Stringent sets, such as the hematopoietic factors SPI1 (n = 27), GATA1 (n = 16) and GATA2 (n = 11), and the hepatic HNF4A (n = 8). This once again suggests conservation of many regulatory interactions, although it is essential to note that this observation is influenced by the limited coverage of ChIP-seq data across biological contexts.

The Elevated collection relaxes the criteria to allow orthologous genes reaching the cut-off in three of the four rankings. This resulted in 3,106 Elevated TR-gene pairs, with 211 of the 216 available TRs having at least one gene in their set (median = 10). TRs with the largest Elevated collection closely overlapped with those having the largest Stringent sets, reinforcing the notion of preserved target genes among these TRs. The Species-specific level encompasses two groups of TRs: those that have ChIP-seq data in both species and those in only one. This is reflected in Fig 5B, where we show the count of reproducible interactions for each group. The left panels display TRs with ChIP-seq in only one species and were thus ineligible for consideration in the Stringent or Elevated tiers. In human, this corresponded to 99 TRs with a median of 11 interactions. TFDP1 led with 93 genes supported by both aggregated single cell coexpression and binding evidence. In mouse, all 89 available TRs were associated with at least one gene (median = 18), with the interferon TR Irf8 having a maximum of 91 genes, including numerous immunity-associated genes such as *Mpeg1*, *Ctss*, *Cd180*, *Xcr1*, and *Trim30a*.

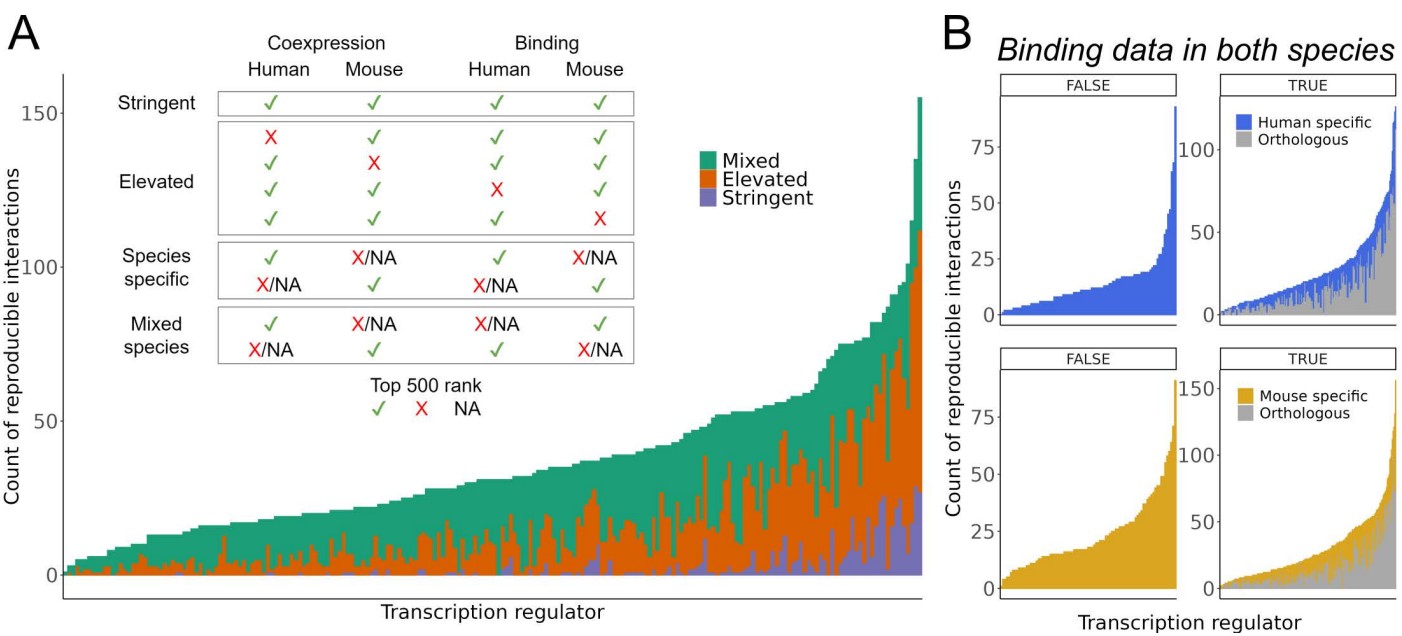

**Fig 5. Count of interactions supported across methods and species.** (A) Inset: criteria used to group interactions into tiers. Bar chart: Count of unique interactions gained in each orthologous tier (Stringent, Elevated, and Mixed-Species) for the 216 TRs with binding data in both species. (B) Count of Species-Specific interactions for 317 TRs in human (top) and 305 TRs in mouse (bottom). TRs are split by those with ChIP-seq data in one species only (left) and thus are ineligible for consideration in the orthologous interactions, and those with ChIP-seq data in both species (right). Grey bars indicate the count of interactions already found in the Stringent and Elevated sets, coloured bars indicate the count of Species-Specific interactions that were gained due to lacking orthologs or because they had elevated ChIP-seq signal in one species and not the other.

## Highlighting ASCL1

We conclude by focusing on ASCL1, emphasizing that this exploration of ASCL1 regulatory targets is just one example made possible by the information we have summarized and made available for community use. In Fig 6A we present the genes in each tier of evidence for ASCL1, along with their curation status from the 39 available ASCL1 targets in the literature corpus. Human ASCL1 was measured in 61 of 120 scRNA-seq datasets, and in mouse 65 of 103. Regarding ASCL1 binding data, there were 10 ChIP-seq datasets in human — largely in cancer cell lines — as well as 10 in mouse, mostly in neuronal and embryonic contexts.

Two genes fit the Stringent criteria used for this report: the literature-curated ASCL1 target and Notch signalling ligand DLL3 [67], and the cell cycling phosphatase CDC25B, which was not in the low-throughput literature collection but is nevertheless discussed elsewhere as a target of ASCL1 [59]. The Elevated set consisted of 26 genes, with 6 narrowly missing the Stringent criteria (indicated by lighter shading in Fig 6A). Among them are well-described and literature-curated ASCL1 targets, such as the Notch effector HES6 [60] and the neuroendocrine regulator INSM1 [68,69]. ASCL1 and INSM1 serve as markers for neuroendocrine tumours, such as for small cell lung carcinoma (SCLC) [70]. Another Elevated ASCL1 gene, CKB, has upregulated expression in both SCLC [71,72] and ASCL1-high atypical teratoid/rhabdoid tumours [73], suggesting an ASCL1 interaction with oncogenic potential across various contexts. We additionally draw attention to the BAF chromatin remodeler BCL7A, for which we found no ASCL1 connection in the literature, and which is also associated with diverse cancers [74,75].

Other Elevated interactions help characterize ASCL1 as a regulator of both neuronal and oligodendrocyte lineages. This includes the cell cycle regulator GADD45G [76], the neuronal

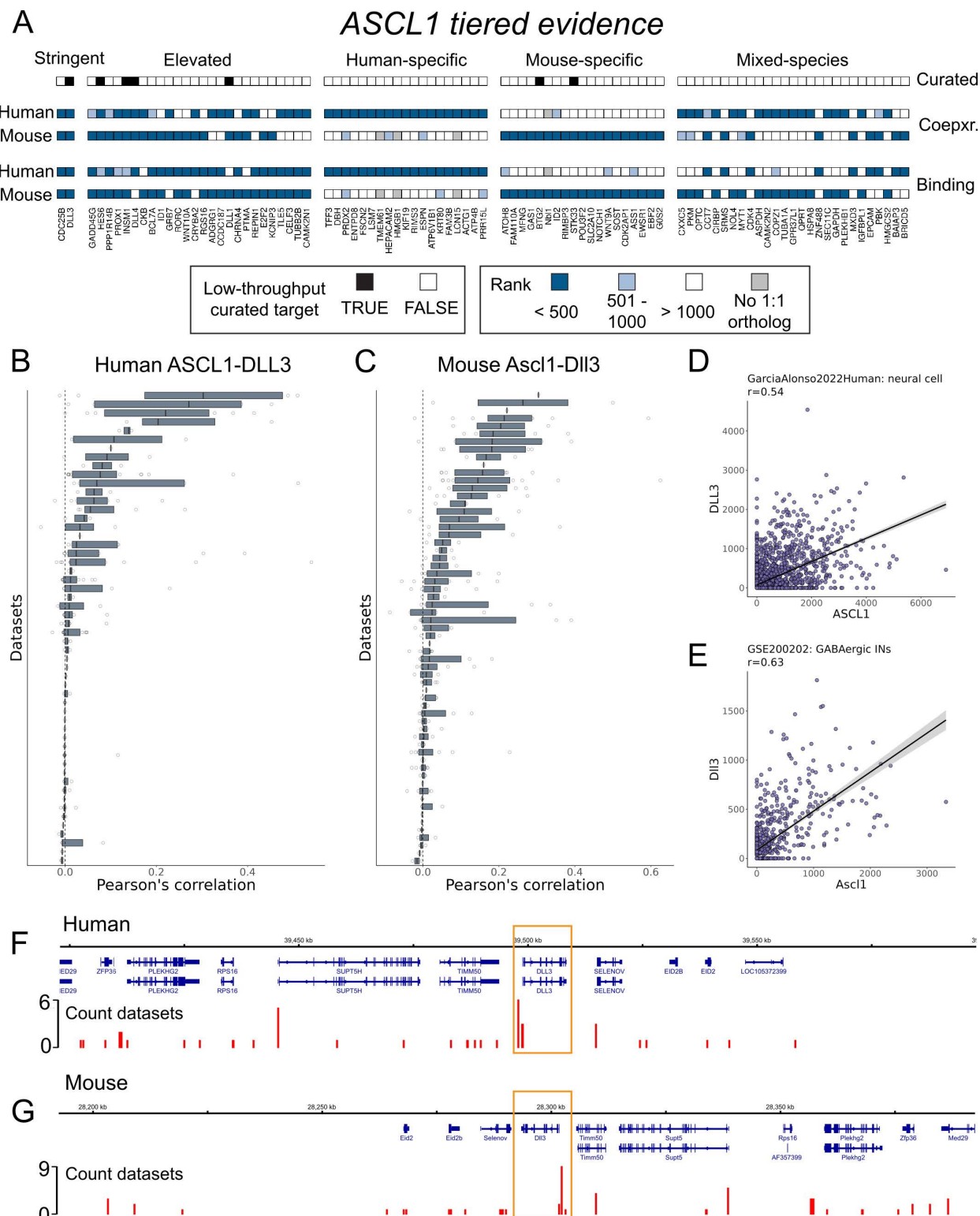

**Fig 6. Reproducible ASCL1 interactions.** (A) Heatmap representing the tiered evidence for ASCL1 candidate targets. (B, C) Distribution of Pearson's correlations for ASCL1-DLL3 in (B) human and (C) mouse, as in Fig 1E–G. (D, E) Scatterplot of the CPM values for ASCL1 and DLL3 for the cells belonging to the cell type that had the highest correlation in the entire corpus for (D) human and (E) mouse. (F, G) Genome track plots centered on DLL3 (yellow boxes) in (F) human and (G) mouse, where the base of the red bars indicates ASCL1 binding regions, and the height indicates the count of ASCL1 ChIP-seq datasets with a peak in the region.

tubulin TUBB2B [77,78], and acetylcholine receptor subunit CHRNA4 [79]. PPP1R14B and ASCL1 expression was used to define a primitive oligodendrocyte progenitor population [80]. We were unable to find (from a low-throughput study or otherwise) a direct connection between ASCL1 and the neuronal adhesion ADGRG1 [81], the cortical-marker and calcium-binding regulator KCNIP3 [82], or the neuronal splicing factor CELF3 [83], although the latter is used as a neuroendocrine marker to characterize ASCL1-high SCLC subtypes [84]. Finally, we highlight REPIN1, an Elevated gene that lacked any ASCL1 connection in the literature that is also generally understudied.

The next tier, of Species-Specific sets, each comprised 19 genes. PRDX2, for example, is a neuronal-enriched mitochondrial gene that has been shown to enhance ASCL1-induced astrocyte-to-neuron reprogramming [85]. HEPACAM2 is another gene implicated in cancer [86,87] that we could not find a direct ASCL1 association in the literature. TMEM61, lacking a 1:1 mouse ortholog, was only eligible for consideration in the Human-specific set, while the reciprocal applied to the mouse Nbl1. Of the 27 genes in the final tier, the Mixed-Species set, we highlight CXXC5. This zinc finger TR was initially characterized as a bone morphogenic-responsive regulator of Wnt signaling in neural stem cells [88], and has been further described as a signal integrator in development and homeostasis with tumour suppressive qualities [89]. These examples collectively illustrate the diverse roles of essential TRs, such as ASCL1, in development and disease.

Lastly, we summarize the compiled evidence for the Notch ligand encoding *DLL3*, a well-established and curated ASCL1 target [67] that was present in the Stringent collection. DLL3 ranked fourth in the ASCL1 coexpression rankings in both species, making it one of ASCL1's most reproducible coexpression partners. Fig 6B and 6C illustrates the distribution of Pearson's correlations for the 238 annotated cell types from 54 human datasets in which ASCL1 and DLL3 were co-measured (275 cell types in 61 datasets for mouse). Notably, despite being one of the most reproducible ASCL1 coexpressions, this association is not universal across all cell types. Fig 6D and 6E shows the scatter plots of the individual cell types in which the greatest correlation was found: in human, annotated as "neural cells" ($r = 0.54$) [90], and in mouse, "GABAergic INs" (interneurons) ($r = 0.63$) [91]. Given the importance of ASCL1 regulation of Notch signalling in neuronal cells [58,59,92], these collective observations support that our resource can still prioritize specific interactions.

In Fig 6F and 6G, we demonstrate the ASCL1-DLL3 binding evidence; DLL3 was ranked 493rd in the human aggregate binding profile and 81st in mouse. In human, this corresponded to 83 discrete bound regions (Methods) within 500Kb of either direction of the DLL3 TSS, and 25 within 100Kb; in mouse 73 regions within 500Kb and also 25 within 100Kb. We calculated which regions were most frequently bound by ASCL1 across datasets, reasoning that this may help prioritize functional ASCL1-DLL3 enhancers (while being cognizant of biasing factors like open promoters). Using the 500Kb cut-off in human, we found that 20 sites were bound in more than one dataset, and that a region approximately 775 base pairs upstream of the DLL3 TSS had a maximum count of 6. In mouse, 28 regions were bound across multiple datasets, with the most frequently bound region (nine of ten datasets) occurring approximately 400 base pairs upstream of the DLL3 TSS.

## Discussion

In this study we pursued two main objectives. First, we aimed to understand the behavior of the meta-analytic strategy of aggregating single cell coexpression networks [1], applying this methodology across a large and broad corpus of scRNA-seq studies. We believe this technique holds great potential in uncovering robust gene coexpression patterns free from the confounding effect of cellular composition. However, before considering specific cell types or

conditions, we sought to calibrate expectations using a large collection of heterogeneous data. This objective aligned with our second aim of identifying reproducible transcription regulator coexpression patterns. We wished to assess how well this information aligns with other lines of regulation evidence, and to provide an organized summary of this information as a community resource (https://doi.org/10.5683/SP3/HJ1B24).

While prior work has nominated TR-target interactions across a large and context-independent corpus of data [23,93,94], to our knowledge ours is the first to do so using a broad range of single cell transcriptomics. Our literature curation benchmark strongly supports the ability of this resource to prioritize curated targets, and we further find numerous examples of reproducible and conserved coexpressed TR-gene partners also supported by ChIP-seq evidence. Collectively, this suggests that this information can help prioritize interactions when direct experimental evidence is lacking. Our benchmarks additionally provide insight into the TRs whose activity is more challenging to uncover, given the considered genomics data (S2 and S3 Tables).

Our workflow prioritizes interactions that are most common across contexts, akin to our prior study [19]. Overall, it is not surprising that the most reproducible relationships tend to relate to processes shared by many cell types. This may be partly a function of expression levels [1], but it is logical that the dynamics of processes like the cell cycle are more readily captured by changing transcript levels. We still find evidence for highly context-specific interactions: as long as there is enough supporting data such patterns can emerge. Conversely, if a TR's activity is highly pleiotropic, our framework will tend to only prioritize the partners shared across data. That we are able to observe reproducible patterns in this heterogenous collection raises our confidence in applying this framework to specific contexts in future work, such as identifying tissue-specific versus global partners for TRs like PAX6.

Repression is more difficult to infer from coexpression than activation, for reasons we discuss in S1 Text. Similarly, differential interactions are more difficult to characterize than those that are reproducible, requiring evidence of absence. While these considerations motivated our focus on the top reproducible coexpression patterns, the data we have organized can help potentiate the discovery of divergent regulatory interactions. Suresh and colleagues [17], for example, used single cell coexpression of human and primate data to nominate both conserved and human-novel coexpression patterns. Given that "TR-rewiring" (differential TR activity) is hypothesized to be a primary driver of phenotypic variation, it would be valuable to assess the degree to which differential coexpression between species in matched contexts can reveal distinct regulatory activity.

Numerous methods have been developed for gene regulatory network reconstruction using single cell coexpression, with multiple benchmarks concluding that no algorithm dominates [95–98]. In particular, McCalla and colleagues [98] emphasized the favorable performance of Pearson's correlation (as used in this study) relative to more complex models. This aligns with observations by Harris et al. [16], who found that aggregating single cell coexpression using the computationally efficient Pearson's correlation provided results that were consistent with alternative similarity metrics [99]. Indeed, we feel that the most important ingredient in the analysis is the aggregation of data because the sparsity of the data is difficult to address otherwise. Our focus on simplistic approaches supports that our conclusions are generalizable to more complex forms of coexpression analysis [1].

We believe that the organized information we provide will be a valuable community resource. Beyond lists of genes plausibly regulated by each TR, the interactions identified in this study can assist studies examining the conservation of regulatory interactions, or the chromatin factors commonly coexpressed with each TR. Highly ranked interactions could be used for benchmarking predictive methods, or further dissected towards our understanding

of the chromatin and sequence features that are characteristic of reproducible interactions. Future work may find it fruitful to construct context-specific aggregations to contrast against this heterogeneous collection, or to further integrate this resource with other lines of regulation evidence, as we did with the ChIP-seq data.

## Supporting information

**S1 Text. Additional commentary.**
(DOCX)

**S1 Table. Included single cell experiment metadata.**
(TSV)

**S2 Table. AUC performances of human TRs in the literature curation benchmark.**
(TSV)

**S3 Table. AUC performances of mouse TRs in the literature curation benchmark.**
(TSV)

**S1 Fig. Gene measurement coverage.** (A) Binary heatmap indicating whether (blue) or not (black) a gene had non-zero counts in at least 20 cells in at least one cell type in a dataset, for 19,213 human protein coding genes and 120 datasets. (B) Mouse: 20,971 protein coding genes and 103 experiments. (C) Schematic of global TR coexpression profile aggregation. Left: Each dataset results in one gene by gene coexpression matrix by aggregating across cell types (schematized in Fig 1C), from which a single gene coexpression profile can be extracted. Right: A given gene's profile (e.g., "TR_a") can be extracted from each dataset. As each profile/dataset will vary in its gene measurement, unmeasured/tied values are imputed to the median value of all of TR_a's profiles before averaging across genes to get TR_a's global coexpression profile.
(PNG)

**S2 Fig. Similarity of negative correlation TR profiles across datasets.** (A) Top panel: Histogram of 1000 iterations of sampling one TR profile from each of 120 human datasets and calculating the average size of the Bottom200 overlap between every pair of sampled profiles, representing a null background setting. Middle panel: Histogram of the average Bottom200 overlap of all dataset pairs for each of 82 ribosomal genes. Bottom panel: Histogram of the average Bottom200 overlap of all dataset pairs for 1,605 human TRs. (B) The average Bottom200 overlap of all human TRs, with the red line indicating the best null overlap. (C,D) Same as in A,B, save for 103 mouse experiments and 1,484 TRs. (E,F) The distribution of (C) Top200 and (D) Top1000 overlaps between every pair of PAX6 and NEUROD6 profiles in human, with ribosomal RPL32, TR E2F8, and a representative null sample included for reference. (G) The distribution of overlaps between TR profiles generated from the same study but using different technology. (H) Each human dataset's Global agreement, a measure averaging how well the TR profiles for a dataset aligned with the global TR profiles. (I-L) Plotting each dataset's Global agreement against its count of (I) genes, (J) cells, (K) cell types, and (L) whether the dataset included any data using the 10X platform.
(PNG)

**S3 Fig. (A, B) Conservation of the consistency of positive (TopK) and negative (BottomK) profiles between mouse and human for 1,228 orthologous TR at (A) K=200 and (B) K=1000.** Each point represents a TR with a one-to-one ortholog between mouse and human. (C, D) Examples of TR profiles with consistent negative, but not positive, profiles at (C) K=200 and (D) K=1000. The TR group considers only TRs whose TopK values were lower

than the typical null TopK value. The Null group shows the range in mean BottomK values of all 1000 shuffled null comparisons.
(PNG)

**S4 Fig. L/S ribosomal gene aggregate profiles prioritize other ribosomal genes in (A) human and (B) mouse.** Each bar represents one of the 82 L/S ribosomal genes with an unambiguous one-to-one ortholog between mouse and human. For each ribosomal gene we aggregated its coexpression profiles and then calculated how many of its top coexpressed partners belonged to the set of 82 ribosomal genes. (C) The top 10 enriched GO terms affiliated with the aggregated coexpression profiles of selected TRs.
(PNG)

**S5 Fig. Literature curated target benchmark.** (A) Aggregating TR coexpression profiles tends to improve recovery of curated targets. Shown are the histograms of the observed AUC quantiles for 451 human and 434 mouse aggregate TR coexpression profiles. For each TR, an AUC is generated for every dataset's ability to recover the given TR's curated targets. This process is repeated for the TR's aggregate profile, which is then compared to all of the individual dataset AUCs. A quantile of 1 indicates that an aggregate profile had a higher AUC (assigned better ranks to curated targets) than all of the individual dataset profiles that compose the aggregate. Black lines correspond to the median aggregate AUC quantile. (B) Distributions of the AUC quantiles for aggregated negative/positive coexpression and binding profiles for the 253 human and 241 mouse TRs that had binding and coexpression data. (C) Histograms of the difference between the raw AUC values between coexpression and binding aggregates. Positive values indicate that coexpression was better able to recover curated targets, negative values indicate binding data was better. (D) Integrating the positive coexpression and binding aggregates (via rank product) tends to increase recovery of curated targets.
(PNG)

**S6 Fig. Comparison of single cell coexpression across species.** (A, B) TRs that are consistent within species tend to be consistent across species. Scatterplots of the $Top_{200}$ overlap between orthologous TRs versus the average $Top_{200}$ between every unique pair of individual TR profiles in (A) human and (B) mouse. Each point is an orthologous TR, the y-axis is a measure of TR coexpression agreement before aggregation within each species, while the x-axis is common to A and B and represents the agreement between aggregate ortholog profiles between species. (C) Scatterplot of the $Bottom_{200}$ ortholog retrieval scores. (D) Scatterplot of the averaged $Bottom_{200}$ and $Top_{200}$ ortholog retrieval scores: the upper right quadrant indicates TRs whose positive and negative coexpression is consistent and specific between species.
(PNG)

## Acknowledgments

We thank Dr. Marine Louarn, Taylor Kim, Joey He, Wilson Tu, and Alice Ma for their work on expanding the literature curation collection of Chu et al. [35]. We are additionally thankful to Mahan Rafieenaini for his assistance in identifying scRNA-seq datasets. The results published here are in whole or in part based on data obtained from the AD Knowledge Portal (https://adknowledgeportal.org), corresponding to Mathys et al. [100] (https://www.synapse.org/Synapse:syn18485175; coded ROSMAP in the Metadata).

## Author contributions

**Conceptualization:** Alexander Morin, Paul Pavlidis.

**Data curation:** Alexander Morin.

**Formal analysis:** Alexander Morin.

**Funding acquisition:** Paul Pavlidis.

**Investigation:** Alexander Morin, Paul Pavlidis.

**Methodology:** Alexander Morin, C. Pan Chu, Paul Pavlidis.

**Project administration:** Paul Pavlidis.

**Resources:** Paul Pavlidis.

**Supervision:** Paul Pavlidis.

**Visualization:** Alexander Morin.

**Writing – original draft:** Alexander Morin.

**Writing – review & editing:** Alexander Morin, C. Pan Chu, Paul Pavlidis.

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
