## [Decision Letter · Decision Letter 0]

10 Dec 2024

PCOMPBIOL-D-24-01939

Identifying Reproducible Transcription Regulator Coexpression Patterns with Single Cell Transcriptomics

PLOS Computational Biology

Dear Dr. Morin,

Thank you for submitting your manuscript to PLOS Computational Biology. After careful consideration, we feel that it has merit but does not fully meet PLOS Computational Biology's publication criteria as it currently stands. Therefore, we invite you to submit a revised version of the manuscript that addresses the points raised during the review process.

Please submit your revised manuscript within 60 days Feb 09 2025 11:59PM. If you will need more time than this to complete your revisions, please reply to this message or contact the journal office at ploscompbiol@plos.org. Please include the following items when submitting your revised manuscript:

We look forward to receiving your revised manuscript.

Kind regards,

Chongzhi Zang

Academic Editor

PLOS Computational Biology

Zhaolei Zhang

Section Editor

PLOS Computational Biology

**Additional Editor Comments :**

While the reviewers accept the overall scientific merits of this study, their concerns and comments in all aspects are valid and should be fully addressed.

**Journal Requirements:**

At this stage, the following Authors/Authors require contributions: Alexander Morin, C. Pan Chu, and Paul Pavlidis. Please ensure that the full contributions of each author are acknowledged in the "Add/Edit/Remove Authors" section of our submission form.

2) We ask that a manuscript source file is provided at Revision. Please upload your manuscript file as a .doc, .docx, .rtf or .tex. If you are providing a .tex file, please upload it under the item type LaTeX Source File and leave your .pdf version as the item type Manuscript.

4) We noticed that you used the phrase 'not shown' in the manuscript. We do not allow these references, as the PLOS data access policy requires that all data be either published with the manuscript or made available in a publicly accessible database. Please amend the supplementary material to include the referenced data or remove the references.

5) Please upload all main figures as separate Figure files in .tif or .eps format. For more information about how to convert and format your figure files please see our guidelines: 

6) We have noticed that you have uploaded Supporting Information files, but you have not included a list of legends. Please add a full list of legends for your Supporting Information files after the references list.

7) Some material included in your submission may be copyrighted. According to PLOS copyright policy, authors who use figures or other material (e.g., graphics, clipart, maps) from another author or copyright holder must demonstrate or obtain permission to publish this material under the Creative Commons Attribution 4.0 International (CC BY 4.0) License used by PLOS journals. Please closely review the details of PLOSu2019s copyright requirements here: PLOS Licenses and Copyright. If you need to request permissions from a copyright holder, you may use PLOS's Copyright Content Permission form.

Potential Copyright Issues:

i) Figures 3A, and 4B. Please confirm whether you drew the images / clip-art within the figure panels by hand. If you did not draw the images, please provide (a) a link to the source of the images or icons and their license / terms of use; or (b) written permission from the copyright holder to publish the images or icons under our CC BY 4.0 license. Alternatively, you may replace the images with open source alternatives. See these open source resources you may use to replace images / clip-art:

8) Please amend your detailed Financial Disclosure statement. This is published with the article. It must therefore be completed in full sentences and contain the exact wording you wish to be published.

1) Please clarify all sources of financial support for your study. List the grants, grant numbers, and organizations that funded your study, including funding received from your institution. Please note that suppliers of material support, including research materials, should be recognized in the Acknowledgements section rather than in the Financial Disclosure

2) State the initials, alongside each funding source, of each author to receive each grant. For example: "This work was supported by the National Institutes of Health (####### to AM; ###### to CJ) and the National Science Foundation (###### to AM)."

3) State what role the funders took in the study. If the funders had no role in your study, please state: "The funders had no role in study design, data collection and analysis, decision to publish, or preparation of the manuscript."

9) Your current Financial Disclosure states, "The author(s) received no specific funding for this work."

However, your funding information on the submission form indicates two funders. Please ensure that the funders and grant numbers match between the Financial Disclosure field and the Funding Information tab in your submission form. Note that the funders must be provided in the same order in both places as well.

Please indicate by return email the full and correct funding information for your study and confirm the order in which funding contributions should appear. Please be sure to indicate whether the funders played any role in the study design, data collection and analysis, decision to publish, or preparation of the manuscript.

**Reviewers' comments:**

Reviewer's Responses to Questions

Reviewer #1: Morin and colleagues have collected >28 million scRNA-seq datasets, and conducted comprehensive analyses to examine TR co-expression profiles at a finer resolution. Their extensive efforts have prioritized reproducible TR interactions across different context and species. The authors also demonstrated that integrating positive coexpression data with ChIP-seq binding data enhances the recovery of curated targets. While I have a generally positive view of the manuscript, several issues should be addressed. The consistency of the single-cell gene expression levels across technologies, conditions needs further attention, additional assessments are required to evaluate the robustness of TR network identification from potential rare cell populations, and there are missed opportunities for more thoroughly characterizing the divergence between human and mice. Furthermore, the authors could consider experimentally validating some novel predicted interactions to enhance the novelty and applicability of this resource. In addition, the following points must be addressed to justify publication in PLOS Computational Biology:

Major comments:

1. Can the authors summarize the proportion of different technologies? Are the TR co-expression profiles consistent between different technologies?

2. The TR network may be in a tissue/organ-specific manner. Could authors report the proportion of different tissues/organs? I wonder if the positively and/or negatively co-expressed gene pairs show any bias towards specific tissues/organs.

3. I wonder if the authors have performed any saturation analysis for each cell type. What’s the minimal number of cells per cell type required for robust TR co-expression network identification?

4. The authors considered TRs that had a minimum of five curated targets. Could authors explain how this cut-off was determined?

5. In Fig 3B, how do authors explain the TRs with significantly lower AUC compared to the median null distribution? I am curious about what kinds of TRs fall in this category?

6. Could authors characterize species-specific TRs? Are they associated with species-specific biological processes, such as immune, and metabolic pathways?

7. Given the different size, shape, and many other factors between human and mouse brain development, I wonder what the divergent TRs are?

8. If possible, I wonder if authors can validate the potential interactions between any novel predicted interactions between ASCL1 and other TRs, such as BCL7A and REPIN1. This evidence may significantly enhance the novelty of this study and the usage of predicted results.

9. In Fig 6F, and G, please provide additional evidence of ASCL1-DLL3 enhancers, by overlapping with ENCODE histone marks or published snATAC-seq datasets in brains?

Minor comments

1. Figure 3B hasn’t been cited.

Reviewer #2: Using coexpression networks at cell-type level and data set basis, the authors first show that similar, reproducible coexpression of transcriptional regulators could be identified in a heterogeneous corpus for both human and mouse. By creating single cell coexpression rankings for each TR, they further demonstrate that in this way literature curated transcriptional targets can be identified, similar to ChIP-seq data. Finally they combined coexpression and ChIP-seq information to identify candidate regulatory interactions supported across methods and species. Overall, the authors created a TR-centered single cell coexpression resource that can be exploited within but also across species.

Major comments

- Line 97 – “Our key aim was to prioritize the genes that are most frequently coexpressed with each TR, hypothesizing that this prioritization can facilitate the identification of direct TR-target interactions.” – coexpression is no synonym for transcriptional TR-target gene interaction, it could also point to physical protein-protein interactions amongst others. Although the authors later give more contextualization to this, I found this statement too strong.

- Line 360 – “The distribution of average Top200 values was highly skewed for TRs.” – What were important TR-TR interactions inferred in this way? Please describe and discuss. Would these be heterodimers or transcriptional regulatory interactions or genetic interactions or …?

- Line 418 – GO enrichment analysis was performed on all aggregated profiles per TR – Why the choice for GO overpresentation analysis and not GSEA?

- Line 488 – “We anticipated that TR ChIP-seq, as a more direct form of regulatory inference, might outperform coexpression.” – Why would this be the case? Literature has shown that binding and expression are two different “modes” defining transcriptional regulation.

- As the authors state themselves, data aggregation is key here. Did the authors validate their data aggregation approach in some way? Did the authors try an alternative data aggregation approach? How much different would the results be when the aggregation was done first across datasets per cell type and then across cell types? Both approaches could be valid and should then be discussed together.

Minor comments

- Orthologs – DIOPT resource v8 – 2019 – is there not a more recent resource, since genome annotation is regularly updated?

- Line 159 – “The resulting rank matrices across cell types were then summed and rank-standardized into the range [0, 1].” – can this be explained by means of an example how this is exactly done?

- Line 192 – “As each dataset-level profile had variable gene measurement, there was variable delineation between the positive coexpression values, the non-measured gene pair ties, and negative coexpression values. Therefore, for a given gene’s set of profiles, we imputed all tied values to the median value before averaging, standardizing the values of non-measured gene pairs.” – can this be explained by means of an example how this is exactly done?

- (See the two comments above) The exact methodological procedure could also be illustrated in more detail in Figure 1.

Reviewer #3: The concept presented in the manuscript is promising, leveraging single-cell RNA-seq data to explore transcription regulator co-expression patterns. However, it is hindered by several critical issues, including methodological shortcomings, insufficient benchmarking, and a lack of clarity in data integration.

The manuscript contains numerous grammatical and syntactical errors that make it difficult to follow. like “We reasoned that evidence of...” , and terms like “most consistent” and “most reproducible” are used without proper definitions or supporting metrics.

The authors calculate gene expression correlation in single-cell level for each cell type using a simplistic approach. However, several potential factors that could bias the co-expression analysis are not adequately addressed, including high dropout rates, unequal number of cells for each cell type , inconsistent annotations across studies.

Many statements in the manuscript lack proper references, such as those discussing OLIG1/2 and neuronal synaptic functionality, or E2F8's role in cytokinesis and chromosomal organization. Providing citations to support these claims is essential to establish their validity and credibility.

Another major issue is that, while the authors highlight the limitations of bulk RNA-seq correlation, they fail to include a comparison with bulk RNA-seq correlation. Considering the numerous potential biases in single-cell correlation, it remains unclear whether single-cell correlation truly offers better performance.

The authors benchmark their predictions against curated datasets; however, they do not provide a systematic comparison and present only a few isolated examples instead.

**Have the authors made all data and (if applicable) computational code underlying the findings in their manuscript fully available?**

Reviewer #1: Yes

Reviewer #2: Yes

Reviewer #3: None

PLOS authors have the option to publish the peer review history of their article (what does this mean? ). If published, this will include your full peer review and any attached files.

**Do you want your identity to be public for this peer review?** For information about this choice, including consent withdrawal, please see our Privacy Policy .

Reviewer #1: No

Reviewer #2: **Yes: ** Vanessa Vermeirssen

Reviewer #3: No

**Figure resubmission:**
---

## [Decision Letter · Decision Letter 1]

28 Feb 2025

PCOMPBIOL-D-24-01939R1

Identifying Reproducible Transcription Regulator Coexpression Patterns with Single Cell Transcriptomics

PLOS Computational Biology

Dear Dr. Morin,

Thank you for submitting your manuscript to PLOS Computational Biology. After careful consideration, we feel that it has merit but does not fully meet PLOS Computational Biology's publication criteria as it currently stands. Therefore, we invite you to submit a revised version of the manuscript that addresses the points raised during the review process.

Please submit your revised manuscript within 30 days Apr 30 2025 11:59PM. If you will need more time than this to complete your revisions, please reply to this message or contact the journal office at ploscompbiol@plos.org. Please include the following items when submitting your revised manuscript:

We look forward to receiving your revised manuscript.

Kind regards,

Chongzhi Zang

Academic Editor

PLOS Computational Biology

Zhaolei Zhang

Section Editor

PLOS Computational Biology

**Journal Requirements:**

1) Please upload all main figures as separate Figure files in .tif or .eps format. For more information about how to convert and format your figure files please see our guidelines:

2) We have noticed that you have uploaded Supporting Information files, but you have not included a list of legends. Please add a full list of legends for your Supporting Information files after the references list.

3) Please ensure that the funders and grant numbers match between the Financial Disclosure field and the Funding Information tab in your submission form. Note that the funders must be provided in the same order in both places as well.

**Reviewers' comments:**

Reviewer's Responses to Questions

**Comments to the Authors:**

Reviewer #1: Overall, I greatly appreciate the authors detailed explanation, substantial analyses, and attempt to compare across different platforms, conduct saturation analysis, explore the negatively correlated TRs. The authors have incorporated more rigorous analyses into both the main figures and supplementary materials and clearly discussed the potential limitations, and revising the manuscript considerably. The other reviewers of this manuscript had some insightful questions that the authors have also addressed. I enjoyed reading this round of revision. However, I would recommend the authors make additional analysis and/or explanation for a few points are detailed here:

1. I appreciate the detailed explanation about rate cell types, please state this potential limitation in the discussion section.

2. I understand it’s unrealistic to process all cell types, it’s challenging to compare across platforms, and the authors would like to focus on the general patterns. Could the author provide additional information or description about how low-quality datasets, platforms, and cell types are excluded from the analyses, making sure they won’t bias to the global pattern as much as possible?

3. I would like to echo reviewer #3’s point. Given the potential bias from co-expression analysis, I would suggest the authors do regression analysis across variables (like platform, cell # per cluster, etc) to correct the expression signals before doing co-expression analysis. Alternatively, I would suggest following my 2nd point, to make sure less low-quality data, and potential bias are included in the analysis.

Reviewer #2: The authors have done a good job in responding to my comments.

Reviewer #3: The authors have answered the comments. No additional concerns

**Have the authors made all data and (if applicable) computational code underlying the findings in their manuscript fully available?**

Reviewer #1: Yes

Reviewer #2: Yes

Reviewer #3: Yes

PLOS authors have the option to publish the peer review history of their article (what does this mean? ). If published, this will include your full peer review and any attached files.

**Do you want your identity to be public for this peer review?** For information about this choice, including consent withdrawal, please see our Privacy Policy .

Reviewer #1: No

Reviewer #2: No

Reviewer #3: No

**Figure resubmission:**
---

## [Editor Report · Decision Letter 2]

13 Mar 2025

Dear Dr. Morin,

We are pleased to inform you that your manuscript 'Identifying Reproducible Transcription Regulator Coexpression Patterns with Single Cell Transcriptomics' has been provisionally accepted for publication in PLOS Computational Biology.

Best regards,

Chongzhi Zang

Academic Editor

PLOS Computational Biology

Zhaolei Zhang

Section Editor

PLOS Computational Biology

---

## [Editor Report · Acceptance letter]

PCOMPBIOL-D-24-01939R2

Identifying Reproducible Transcription Regulator Coexpression Patterns with Single Cell Transcriptomics

Dear Dr Pavlidis,

I am pleased to inform you that your manuscript has been formally accepted for publication in PLOS Computational Biology. Your manuscript is now with our production department and you will be notified of the publication date in due course.

With kind regards,

Anita Estes
